# Factors Associated with Prolonged Antibiotic Therapy in Neonates with Suspected Early-Onset Sepsis

**DOI:** 10.3390/antibiotics13050388

**Published:** 2024-04-25

**Authors:** Bo M. van der Weijden, Jolien R. van Dorth, Niek B. Achten, Frans B. Plötz

**Affiliations:** 1Department of Paediatrics, Tergooi MC, Laan van Tergooi 2, 1212 VG Hilversum, The Netherlands; 2Department of Paediatrics, Amsterdam UMC, Emma Children’s Hospital, Meibergdreef 9, 1105 AZ Amsterdam, The Netherlands; 3Department of Paediatrics, Erasmus MC, Sophia Children’s Hospital, Wytemaweg 80, 3015 CN Rotterdam, The Netherlands

**Keywords:** anti-bacterial agent, antimicrobial stewardship, blood culture, neonatal sepsis, newborn

## Abstract

Early-onset sepsis (EOS) is a rare but profoundly serious bacterial infection. Neonates at risk of EOS are often treated with antibiotics. The start of empiric antibiotic therapy can successfully be reduced by the implementation of the EOS calculator. However, once started, antibiotic therapy is often continued despite a negative blood culture. To decrease the burden of antibiotic therapy, it is necessary to know whether the clinician’s reasons are based on objective factors. Therefore, we performed a retrospective single-centre cohort study to identify the factors associated with prolongation of antibiotic therapy in neonates with suspected EOS but a negative blood culture. Maternal, clinical, and laboratory data of neonates with a gestational age of ≥32 weeks, admitted between January 2019 and June 2021, were collected. Among neonates with a negative blood culture, we compared neonates with prolonged (≥3 days) to neonates with discontinued (<3 days) antibiotic therapy. The clinician’s reported reasons for prolonging therapy were explored. Blood cultures were positive in 4/146 (2.7%), negative in 131/146 (89.7%), and not obtained in 11/146 (7.5%) of the neonates. The incidence of EOS was 0.7 per 1000 neonates. Of the 131 neonates with a negative blood culture, 47 neonates (35.9%) received prolonged antibiotic therapy. In the prolonged group, the mean gestational age was higher (38.9 versus 36.8 weeks), and spontaneous preterm birth was less prevalent (21.3% versus 53.6%). Prolonged treatment was associated with late onset of respiratory distress, respiratory rate, hypoxia, apnoea and bradycardia, pale appearance, behavioural change, and elevated CRP levels. The most reported reasons were clinical appearance (38.3%), elevated CRP levels (36.2%), and skin colour (10.6%). Prolonging empiric antibiotic therapy despite a negative blood culture is common in suspected EOS. Clinical signs associated with prolongation are uncommon and the reported reasons for prolongation contain subjective assessments and arbitrary interpretations that are not supported by the guideline recommendations as arguments for prolonged therapy.

## 1. Introduction

Early-onset sepsis (EOS) is a rare but profoundly serious bacterial infection, which occurs within 72 h after birth, and is confirmed by a positive blood culture or cerebrospinal fluid culture. The most common bacteria causing EOS are Group B Streptococci (GBS) and Escherichia coli [1,2]. The majority of guidelines advocate the use of a beta-lactam antibiotic (typically benzylpenicillin or ampicillin) in conjunction with an aminoglycoside (most commonly gentamicin) for the empirical treatment of all cases of EOS [3]. The incidence of EOS is between 0.13 and 1.45 per 1000 live births [4]. However, up to a 58 times higher number of neonates receive antibiotic therapy for suspected EOS compared to the number of neonates with a positive blood culture [4]. Adverse consequences of antibiotic overtreatment are increasingly recognized [5]. In the short term, it leads to the separation of parents and newborns, invasive painful procedures (with possible long-term consequences for brain development), and unnecessary hospitalizations with associated costs. In the longer term, there are significant disadvantages for the microbiome and immune system, found to be associated with obesity [6], eczema, asthma [7,8], and diabetes later in life [9]. In addition, at the population level, the overuse of antibiotics leads to resistant infections that are difficult to treat. 

The start of empiric antibiotic therapy can successfully be reduced by the implementation of the EOS calculator, which calculates an individual EOS risk in neonates with a gestational age of ≥34 weeks [10]. However, once started, antibiotic therapy is prolonged for more than three days in about 30% of neonates despite a negative blood culture [11]. The reported reasons for continuing antibiotic therapy include doubts about the quality of the blood culture [12]. Clinical signs at the start of antibiotic therapy or elevated C-reactive protein (CRP) levels are also associated with a prolonged duration of antibiotic therapy in both term and preterm neonates with suspected EOS and a negative blood culture [13]. Clinicians seem to be facing difficulties in finding a balance between the adequate treatment of EOS and the unnecessary use of antibiotics. Current guidelines may not provide sufficient tools to achieve this balance. 

To decrease the burden of antibiotic therapy in neonates, it is necessary to investigate the motivation and reasons for clinicians to continue antibiotic therapy despite a negative blood culture and whether these reasons are based on objective factors. We, therefore, compared maternal, clinical, and laboratory data of neonates with a negative blood culture with prolonged (≥3 days) antibiotic therapy to those with discontinued (<3 days) antibiotic therapy. Secondary outcomes included reported motivation and reason for clinicians to continue antibiotic therapy.

## 2. Results

In total, 5522 neonates were born during the study period, of whom 146 (2.6%) received antibiotic therapy for suspected EOS. Blood-culture results were negative in 131/146 (89.7%), positive in 4/146 (2.7%), and not obtained in 11/146 (7.5%) of the neonates, respectively. Lumbar punctures were obtained in 3/146 (2.1%) of the neonates, all (100%) of which were negative. The incidence of EOS was 4/5522 (0.7 per 1000 neonates). Of the 131 neonates with a negative blood culture, 47 (35.9%) received prolonged antibiotic therapy (Table 1).

### 2.1. Maternal Risk Factors at the Start of Antibiotic Therapy

The mean gestational age was higher in neonates with prolonged antibiotic therapy, namely 38.9 (SD 2.5) versus 36.8 weeks (SD 3.2), respectively (Table 1). Spontaneous preterm birth was less prevalent in neonates with prolonged antibiotic therapy (10/47 (21.3%) versus 45/84 (53.6%)).

### 2.2. Neonatal Risk Factors at Start of Antibiotic Therapy

Neonates who received prolonged antibiotic therapy more often had a late onset of respiratory distress (more than 4 h postpartum), a higher respiratory rate, and a pale skin colour (Table 2). Respiratory distress and a pale skin colour were present in less than half of the neonates. The prolonged group had more signs of hypoxia and supplemental, low flow, oxygen therapy. Continuous positive airway pressure (CPAP) support was less present, namely in 17/47 neonates (36.2%) versus 44/84 neonates (52.4%).

### 2.3. Neonatal Clinical Condition at Day 3 of Treatment

Differences in behavioural change, presence of apnoea/bradycardia, hypoxia, a pale skin colour, and less responsiveness were significantly more present in the prolonged group (Table 3).

### 2.4. Laboratory Results

Both the first and the second CRP levels were higher in the group with prolonged antibiotic therapy. The median first CRP level was 10.5 mg/dL (IQR 42.0) versus 1.0 mg/dL (IQR 0.0), respectively, and the second CRP level was 24.0 mg/dL (IOR 43.7) versus 2.0 mg/dL (IQR 6.0), respectively. The median time between birth and the first CRP measurement was 9.0 h (IQR 16.5) in the group with prolonged antibiotic therapy compared to 2.0 h (IQR 8.3) in the discontinued group (*p* < 0.05). The median time between birth and the second CRP measurement was comparable: 32.0 h (IQR 27.5) versus 32.0 h (IQR 21.0), respectively (Table 4).

### 2.5. Calculated EOS Risk

In 91/131 neonates (69.5%), with a gestational age of ≥34 weeks, the EOS calculator was used to determine the EOS risk. Prolonged antibiotic therapy was received in 36/91 of the neonates (39.6%). The EOS calculator recommended significantly fewer starts of antibiotic therapy immediately after the first clinical evaluation in the prolonged group, namely 47.2% versus 70.9%, respectively (Table 5). Also, the change in the neonatal clinical condition over time, which resulted in higher EOS risk and a change in the EOS calculator recommendation, was associated with the prolongation of therapy (Table 5).

### 2.6. Motivation and Reasons to Continue Antibiotic Therapy

Clinician’s most reported reasons to prolong antibiotic therapy included clinical appearance (18/47 (38.3%)), elevated CRP levels (17/47 (36.2%)), and skin colour (5/47 (10.6%)). In 4/47 of neonates (8.5%), a prolonged antibiotic therapy course was set at the start of antibiotic therapy, independent of the culture results. In 25.5%, the reasons for prolonging antibiotic therapy were not noted in the medical record. A complete overview of the reported reasons is available in Appendix A. 

## 3. Material and Methods

### 3.1. Study Design and Participants

A retrospective cohort study was performed at Tergooi MC Blaricum, a non-academic hospital in the Netherlands. The annual total number of deliveries at our hospital was requested by the obstetrics department. The hospital pharmacy provided a list of all neonates, admitted between January 2019 and June 2021, who received antibiotics. Neonates were included if they had (1) a postmenstrual age of 32 weeks or more and (2) antibiotics were administered for (suspected) EOS. Suspected EOS was defined as the presence of at least one of the following criteria: spontaneous prematurity (≥32 weeks), maternal intrapartum fever, long-term ruptured membranes (>18 h for premature labour and >24 h for term labour), GBS positive status of the mother, maternal antibiotic use, and/or suspected neonatal infection based on the neonatal clinical condition within the first 72 h after birth. EOS was defined as an infection in the first 72 h after birth confirmed by a positive blood culture and/or cerebrospinal fluid culture. Excluded were neonates with (a suspicion of) chromosomal abnormalities or serious congenital anomalies and neonates born in another hospital. 

### 3.2. Antibiotic Treatment Guidelines

For neonates between 32 and 34 weeks of gestational age, the Dutch guideline was applied for the management of (suspected) EOS. The Dutch guideline uses eight maternal risk factors and 15 neonatal clinical symptoms to assess the risk of EOS, and a flowchart guides the clinicians on whether to start antibiotic therapy (Appendix A) [14]. For neonates born after a gestational age of more than 34 weeks, the Dutch guideline or the EOS calculator was used to decide on the start of antibiotics, depending on the clinician’s preference. If antibiotic therapy was started, it was administered for at least 48–72 h. The Dutch guideline recommends discontinuing antibiotic therapy after 36–48 h in case of (1) negative blood culture(s), (2) low initial EOS suspicion, and (3) the absence of clinical symptoms for suspected infection in combination with reassuring laboratory tests. The EOS calculator has recommendations on the start of antibiotics and not on the duration of therapy.

### 3.3. Data Collection

The following data were collected from the electronic health record HiX, version 6.2 (ChipSoft B.V., Amsterdam, The Netherlands): maternal risk factors and neonatal clinical symptoms according to the Dutch guideline and the EOS calculator, risks and recommendations according to the EOS calculator for neonates with a gestational age of more than 34 weeks, blood-culture results and cerebrospinal fluid culture results, CRP levels, the timing of the start of antibiotic therapy, the duration of antibiotic therapy, and decision-making on the duration of antibiotic therapy. The presence of risk factors and neonatal clinical signs was collected at the start of antibiotic therapy and when blood culture results were considered negative (72 h). If neonatal risk factors were not reported, these were considered absent. Finally, reasons for continuing antibiotic therapy according to clinicians were collected. When a neonate was transferred to another hospital, data on the duration of antibiotic therapy, clinical signs, and laboratory results were requested at the hospital to which the neonate was transferred or obtained from the medical correspondence. Data were collected from digital patient records and were processed in Castor Electronic Data Capture (version 2023.2.2.2).

### 3.4. Study Variables

The primary outcome was prolonged antibiotic therapy, defined as empiric antibiotic therapy started for EOS suspicion and continued uninterrupted for longer than or equal to three days. Secondary outcomes included reported motivation and the reasons for clinicians to continue antibiotic therapy despite negative blood-culture results.

### 3.5. Data Analysis

Statistical analyses were performed using SPSS Statistics, version 28 (IBM Corp., Armonk, NY, USA). Categorical variables were compared using a chi-square test, and the results were presented as the number of cases with percentages. For the comparison of numerical variables, two sample *t*-tests were used for normally distributed data, and the results were expressed as the mean with standard deviation (SD). Mann–Whitney U-tests were used for non-normally distributed data, and the results were presented as the median with an interquartile range. A *p*-value of <0.05 was considered statistically significant.

### 3.6. Ethical Statement

This study was approved by the board of Tergooi MC Blaricum (study number: 21.52). The study was not subject to the Dutch Medical Research Involving Human Subjects Act. All procedures involving human participants were performed in accordance with the principles of the Declaration of Helsinki.

## 4. Discussion

In this cohort, 2.6% of the neonates received empiric antibiotic therapy for suspected EOS, of whom 35.9% received antibiotic therapy for at least three days despite negative blood cultures. Several significant differences in neonatal clinical risk factors were observed between these groups at the start of antibiotic therapy and when blood cultures were considered negative. However, those risk factors were rare in both groups. The clinician’s most reported reasons for prolonging antibiotic therapy included neonatal clinical appearance, the elevation of the CRP levels, and skin colour. Hence, the factors associated with prolonged treatment were either uncommon or subject to interpretation and, thus, do not justify the amount of prolonged antibiotic therapy.

Most international guidelines recommend discontinuing empiric antibiotic therapy for suspected EOS in case there is a low initial suspicion of EOS at the start of antibiotic therapy and the clinical condition and laboratory results are reassuring [1,14,15,16,17]. Our results show that the interpretation of these criteria is not straightforward. Moreover, clear guidance on the duration of antibiotics in the absence of these criteria in the context of a negative blood culture is lacking. These difficulties are known to lead to unnecessarily prolonged antibiotic therapy in case the blood culture is negative [5,12]. 

Interpretation of the risk of EOS at the start of antibiotic therapy is difficult [13]. In this study, we examined objective maternal and neonatal risk factors based on the Dutch guideline. Maternal risk factors were comparable between both groups, except that full-term neonates received more often prolonged therapy compared to spontaneously premature-born neonates. Several statistically significant differences in neonatal clinical signs were observed; though, not all of them may be considered clinically relevant. Most of these clinical signs, for example, late-onset respiratory distress, higher respiratory rate, and pale appearance, were present in less than half of the neonates, and all decreased over time. Those symptoms may be related to EOS or other more common conditions, for example, persistent pulmonary hypertension of the newborn or prematurity, present with the same features. Furthermore, the interpretation of neonatal clinical signs to predict the likelihood of EOS varies considerably between physicians. It is remarkable that, in our clinical practice, to overcome these discussions, the duration of antibiotic therapy is often decided at the start of antibiotic therapy, independent of the culture results. Our findings show that this is subjective, and we advocate that this approach should be abandoned.

The interpretation of elevated CRP levels and their sensitivity to EOS are also frequently a source of debate on continuing antibiotic therapy despite negative blood cultures and improved clinical conditions. Unfortunately, elevated CRP levels are often considered to have a high positive predictive value for EOS, which is simply not true [1]. Elevated CRP levels were an important reported reason to continue antibiotic therapy, which was confirmed by the significant differences in CRP levels between both groups. However, absolute levels were elevated only moderately, given that considerable variations in CRP levels, up to at least 40–50 mg/L but also >100 mg/L in non-infected neonates, have been seen [18,19]. In addition, the current threshold of 10 mg/L is excessively low and is associated with more laboratory investigations and longer admission [20]. In Sweden, a maximum CRP level of 80 mg/L is used, if it decreases by at least 50% within three days postpartum. This strategy significantly decreased the duration of antibiotic therapy and hospital stay and, hence, reduced healthcare costs, with no reinfection in a cohort of term neonates [19]. PCT-guided decision making is an alternative, which proved to safely reduce the duration of antibiotic therapy in late preterm and term neonates [21].

We hypothesized that the individual calculated EOS risk at the start of antibiotic therapy correlated with the duration of therapy. The EOS calculator estimates the EOS risk based on five objective maternal and four clinical neonatal risk factors. Surprisingly, higher risk estimates were not associated with the prolonging of antibiotic therapy. This is explained by the need for respiratory support in the first days after birth due to, for example, respiratory distress syndrome or transient tachypnoea in the neonate. The need for respiratory support increases the risk estimation by the EOS calculator, regardless of the cause for this need. This highlights that the EOS calculator was developed for decision making regarding the start of empiric antibiotic therapy and not for the duration of therapy.

The strengths of our study include the robust analysis of all maternal and neonatal risk factors that are part of the Dutch national guideline. Furthermore, we also looked at two time points, at the start of antibiotic therapy and when the blood-culture results were available to decide on the continuation of antibiotic therapy. Since our study was retrospective, observer bias to report reasons by clinicians to prolong therapy was prevented. We acknowledge the limitations of our study. The study design was single centred, which means that the results may be different for other centres. However, our results are in line with a large cross-sectional study of 757,979 neonates born in 13 networks from 11 countries, in whom 2.86% received antibiotics during the first postnatal week, with a median (IQR) treatment duration of 4 days (3–6) for those without EOS [4]. It is a retrospective study and data collection was limited with respect to vital parameters and the reported reasons by clinicians for continuing antibiotic therapy. The reasons were not always reported, were noted in general terms, or were difficult to find in the patient records.

In clinical practice, cases with prolonged antibiotic therapy are often considered ‘culture-negative’ EOS [5]. Common misconceptions leading to the conclusion of a blood culture being false negative include perceived low sensitivity, assumed interference by intrapartum antibiotic prophylaxis, and assumed inferiority to the clinician’s own judgement [12]. Consequently, culture-negative EOS is a frequent diagnosis, which results in prolonged treatment, even though its existence is uncertain. Our study demonstrates that objective criteria are needed to guide clinicians on whether and for what duration prolonged antibiotic treatment is appropriate.

## 5. Conclusions

Prolonging empiric antibiotic therapy despite a negative blood culture is common in suspected EOS. In more than one-third of neonates with a negative blood culture, antibiotic therapy was administered for >72 h, predominantly because of clinical signs of infection and elevated inflammation markers. However, these factors and the reported reasons are uncommon and contain subjective assessments and arbitrary interpretations unsupported by the guideline recommendations. International guidelines are warranted to guide clinicians on when to prolong antibiotic therapy in case of a negative blood culture. We advocate that this only applies to a small number of neonates and that antibiotic therapy can be stopped in the majority of cases with a negative blood culture. This will help to further reduce unnecessary antibiotic exposure.

## Figures and Tables

**Table 1 antibiotics-13-00388-t001:** Comparison of maternal risk factors * between neonates with a negative blood culture in whom antibiotic therapy was discontinued or prolonged after 72 h.

	Discontinued AB (*n* = 84)	Prolonged AB (*n* = 47)	Total (*n* = 131)	*p*-Value
Gender, male (n (%))	42 (50.0)	22 (46.8)	64 (48.9)	0.726
Gestational age (mean (SD))	36.8 (3.2)	38.9 (2.5)	37.6 (3.1)	**0.000**
Apgar score after 1 min (median (IQR))	9.0 (2.0) ^a^	8.0 (3.0) ^b^	9.0 (2.0) ^c^	0.217
Apgar score after 5 min (median (IQR))	10.0 (2.0) ^b^	9.0 (2.0) ^b^	9.0 (2.0) ^a^	0.231
Apgar score after 10 min (median (IQR))	10.0 (1.0) ^b^	10.0 (2.0) ^b^	10 (1.0) ^a^	0.277
Type of maternal intrapartum antibiotics (n (%)) ^a^				
- No maternal antibiotics or <2 h antepartum	66 (78.6)	39 (83.0)	105 (80.2)	0.544
- GBS-specific antibiotics ≥ 2 h antepartum	5 (6.0)	2 (4.3)	7 (5.3)	0.679
- Broad spectrum antibiotics 2–4 h antepartum	5 (6.0)	4 (8.5)	9 (6.9)	0.579
- Broad spectrum antibiotics > 4 h antepartum	7 (8.3)	1 (2.1)	8 (6.1)	0.155
Twin with EOS (n (%))	0 (0) ^a^	1 (2.1) ^a^	1 (0.8) ^d^	0.175
Invasive GBS in a previous child (n (%))	1 (1.2) ^e^	0 (0) ^f^	1 (0.8) ^g^	0.436
Obtained maternal GBS status (n (%))	55 (65.5) ^b^	23 (48.9) ^b^	78 (59.5) ^a^	0.070
Maternal GBS positivity (n (%))	10 (11.9)	6 (12.8)	16 (12.2)	0.430
Known maternal GBS status at start of treatment (n (%))	14 (16.7) ^h^	3 (6.4) ^i^	17 (13.0) ^j^	0.229
Duration of ROM (mean (SD))	32.8 (50.4) ^a^	22.6 (21.3) ^a^	29.2 (42.8) ^d^	0.202
Prolonged ROM (n (%)) ^f^	32 (38.1)	17 (36.2)	49 (37.4)	0.827
Spontaneous preterm birth (n (%))	45 (53.6)	10 (21.3)	55 (42.0)	**0.000**
Highest maternal temperature (mean (SD)) ^k^	37.9 (0.8)	37.8 (0.7)	37.9 (0.7)	0.534
Maternal intrapartum fever (n (%))	21 (25.0)	14 (29.8)	35 (26.7)	0.553
Maternal sepsis (n (%))	1 (1.2) ^a^	0 (0) ^b^	1 (0.8) ^c^	0.452
Maternal chorioamnionitis (n (%))	14 (16.7) ^a^	12 (25.5) ^b^	26 (19.8) ^c^	0.224
Maternal bacteriuria or urinary tract infection (n (%))	9 (10.7)	9 (19.1)	18 (13.7)	0.179

* Based on the Dutch guideline [14]. Bold *p*-Values are considered statistically significant. Abbreviations: EOS, early-onset sepsis; GBS, Group B streptococcus; IQR, interquartile range; ROM, rupture of membranes; SD, standard deviation. ^a^ 2 missing, ^b^ 1 missing, ^c^ 3 missing, ^d^ 4 missing, ^e^ 29 missing, ^f^ 14 missing, ^g^ 43 missing, ^h^ 31 missing, ^i^ 25 missing, ^j^ 56 missing, ^k^ 24 missing.

**Table 2 antibiotics-13-00388-t002:** Comparison of neonatal risk factors * at the start of antibiotic therapy between neonates with a negative blood culture, in whom antibiotic therapy was discontinued or prolonged after 72 h.

	Discontinued AB (*n* = 84)	Prolonged AB (*n* = 47)	Total (*n* = 131)	*p*-Value
Respiratory distress > 4 h postpartum (n (%))	13 (15.5)	18 (38.3)	31 (23.7)	**0.003**
Ventilation in a preterm neonate (n (%))	3 (3.6)	0 (0)	3 (2.3)	0.425
Shock (n (%))	2 (2.4)	1 (2.1)	3 (2.3)	0.926
Behavioural change (n (%))	49 (58.3)	35 (74.5)	84 (64.1)	0.065
Feeding difficulties (n (%))	14 (16.7)	6 (12.8)	20 (15.3)	0.552
Apnoea/bradycardia (n (%))	25 (29.8)	12 (25.5)	37 (28.2)	0.606
Respiratory distress (n (%))	63 (75.0)	41 (87.2)	104 (79.4)	0.097
Hypoxia (n (%))	37 (44.0)	31 (66.0)	68 (51.9)	**0.016**
Encephalopathy (n (%))	2 (2.4)	1 (2.1)	3 (2.3)	0.926
Need for CPR (n (%))	2 (2.4)	0 (0)	2 (1.5)	0.286
Ventilation in a term neonate (n (%))	1 (1.2)	2 (4.3)	3 (2.3)	0.261
PPHN (n (%))	3 (3.6)	2 (4.3)	5 (3.8)	0.845
Temperature instability (n (%))	16 (19.0)	12 (25.5)	28 (21.4)	0.385
Local signs of infection (n (%))	2 (2.4)	0 (0)	2 (1.5)	0.286
Heart rate (mean (SD))	148 (34)	139 (54)	145 (41) ^a^	0.352
Temperature (mean (SD))	36.9 (0.9)	36.6 (0.7)	36.8 (0.9) ^b^	0.181
Respiratory rate (mean (SD)	67 (16.7)	76 (21.0)	70 (19) ^c^	**0.031**
Respiratory support (n (%))	54 (64.3)	28 (59.6)	82 (62.6)	0.593
Type of respiratory support **				
- CPAP (n (%))	44 (81.5)	17 (60.7)	61 (74.4)	**0.041**
- High flow (n (%))	1 (1.9)	0 (0)	1 (1.2)	0.469
- Low flow (n (%))	9 (16.7)	11 (39.3)	20 (24.4)	**0.024**
Supplemental oxygen (n (%))	28 (51.9)	22 (78.6)	50 (61.0) ^d^	**0.029**
Percentage of supplemental oxygen (mean (SD))	58 (32)	62 (29)	60 (30)	0.709
Vasopressors (n (%))	1 (1.2)	1 (2.1)	2 (1.5)	0.675
Skin colour **				
- No abnormalities (n (%))	63 (75.0)	18 (38.3)	81 (61.8)	**0.000**
- Jaundice (n (%))	2 (2.4)	3 (6.4)	5 (3.8)	0.252
- Pale (n (%))	13 (15.5)	21 (44.7)	34 (26.0)	**0.000**
- Red (n (%))	1 (1.2)	1 (2.1)	2 (1.5)	0.675
- Blue (n (%))	2 (2.4)	0 (0)	2 (1.5)	0.286
- Gray (n (%))	3 (3.6)	4 (8.5)	7 (5.3)	0.228
Neurological state **				
- No abnormalities (n (%))	64 (76.2)	24 (51.1)	88 (67.2)	**0.003**
- Irritable (n (%))	13 (15.5)	13 (27.7)	26 (19.8)	0.094
- Less responsive (n (%))	6 (7.1)	6 (12.8)	12 (9.2)	0.285
- Irritable and less responsive (n (%))	1 (1.2)	4 (8.5)	5 (3.8)	**0.036**
Immediate start of antibiotic therapy (n (%))	59 (70.2)	20 (42.6)	79 (60.3)	**0.002**

* Based on the Dutch guidelines [14]. Bold *p*-Values are considered statistically significant. Hypoxia is defined as either an oxygen saturation < 92%, central cyanosis, or the need for supplemental oxygen to maintain a saturation ≥ 92%. ** Not all percentages add up to 100 due to rounding. Abbreviations: CPAP, continuous positive airway pressure; CPR, cardiopulmonary resuscitation; PPHN, persistent pulmonary hypertension of the newborn; SD, standard deviation. ^a^ 47 missing, ^b^ 50 missing, ^c^ 30 missing, ^d^ 2 missing.

**Table 3 antibiotics-13-00388-t003:** Comparison of neonatal risk factors * when blood-culture results were considered negative (72 h).

	Discontinued AB (*n* = 84)	Prolonged AB (*n* = 47)	Total (*n* = 131)	*p*-Value
Respiratory distress > 4 h postpartum (n (%))	1 (1.2) ^a^	2 (4.3)	3 (2.3) ^a^	0.276
Seizures (n (%))	0 (0) ^a^	1 (2.1)	1 (0.8) ^a^	0.188
Ventilation in a preterm neonate (n (%))	1 (1.2)	2 (4.3)	3 (2.3)	0.606
Shock (n (%))	2 (2.4) ^a^	3 (6.4)	5 (3.8) ^a^	0.271
Behavioural change (n (%))	1 (1.2) ^a^	5 (10.6)	6 (4.6) ^a^	**0.015**
Feeding difficulties (n (%))	9 (10.7) ^a^	8 (17.0)	17 (13.0) ^a^	0.342
Apnoea/bradycardia (n (%))	0 (0) ^a^	3 (6.4)	3 (2.3) ^a^	**0.021**
Respiratory distress (n (%))	20 (23.8) ^b^	14 (29.8) ^c^	34 (26.0) ^d^	0.487
Hypoxia (n (%))	8 (9.5) ^a^	11 (23.4) ^e^	19 (14.5) ^f^	**0.033**
Encephalopathy (n (%))	1 (1.2) ^a^	1 (2.1)	2 (1.5) ^a^	0.695
Ventilation in a term neonate (n (%))	2 (2.4)	0 (0)	2 (1.5)	0.507
PPHN (n (%))	1 (1.2) ^a^	1 (2.1)	2 (1.5) ^a^	0.695
Temperature instability (n (%))	1 (1.2) ^a^	0 (0)	1 (0.8) ^a^	0.444
Local signs of infection (n (%))	1 (1.2) ^a^	0 (0)	1 (0.8) ^a^	0.444
Heart rate (mean (SD))	134 (21) ^g^	131 (16) ^h^	133 (19) ^i^	0.385
Temperature (mean (SD))	37.1 (0.3) ^j^	37.1 (0.3) ^f^	37.1 (0.3) ^k^	0.125
Respiratory rate (mean (SD))	46 (13) ^g^	50 (16) ^h^	48 (14) ^i^	0.149
Respiratory support (n (%))	13 (15.5) ^a^	13 (27.7)	26 (19.8) ^a^	0.116
Type of respiratory support **				
- CPAP (n (%))	2 (15.4)	2 (15.4)	4 (15.4)	1.000
- High flow (n (%))	4 (30.8)	1 (7.7)	5 (19.2)	0.135
- Low flow (n (%))	4 (30.8)	7 (53.8)	11 (42.3)	0.234
- Endotracheal intubation (n (%))	3 (23.1)	3 (23.1)	6 (23.1)	1.000
Supplemental oxygen (n (%))	7 (53.8)	5 (38.5) ^c^	12 (46.2) ^c^	0.682
Percentage of supplemental oxygen (mean (SD))	42 (33) ^c^	64 (42) ^e^	52 (36)	0.408
Vasopressors (n (%))	2 (2.4) ^a^	3 (6.4)	5 (3.8) ^a^	0.271
Skin color				
- No abnormalities (n (%))	50 (59.5) ^c^	27 (57.4)	77 (58.8) ^c^	0.694
- Jaundice (n (%))	39 (46.4) ^c^	19 (40.4)	58 (44.3) ^c^	0.433
- Pale (n (%))	2 (2.4) ^c^	9 (19.1)	11 (8.4) ^c^	**0.001**
- Red (n (%))	1 (1.2) ^c^	0 (0)	1 (0.8) ^c^	0.447
- Gray (n (%))	0 (0) ^c^	1 (2.1)	1 (0.8) ^c^	0.185
Neurological state				
- No abnormalities (n (%))	79 (94.0) ^e^	41 (87.2)	120 (91.6) ^e^	0.102
- Irritable (n (%))	4 (4.8) ^e^	3 (6.4)	7 (5.3) ^e^	0.704
- Less responsive (n (%))	0 (0) ^e^	3 (6.4)	3 (2.3) ^e^	**0.020**

* Based on the Dutch guidelines [14]. Bold *p*-Values are considered statistically significant. Hypoxia is defined as either an oxygen saturation < 92%, central cyanosis, or the need for supplemental oxygen to maintain a saturation ≥ 92%. ** Not all percentages add up to 100 due to rounding. Abbreviations: CPAP, continuous positive airway pressure; PPHN, persistent pulmonary hypertension of the newborn; SD, standard deviation. ^a^ 3 missing, ^b^ 5 missing, ^c^ 2 missing, ^d^ 7 missing, ^e^ 1 missing, ^f^ 4 missing, ^g^ 25 missing, ^h^ 9 missing, ^i^ 34 missing, ^j^ 10 missing, ^k^ 14 missing.

**Table 4 antibiotics-13-00388-t004:** C-reactive protein levels (in mg/dL) and time (in hours) to start antibiotic therapy in newborns with a negative blood culture.

	Discontinued AB (*n* = 84)	Prolonged AB (*n* = 47)	Total (*n* = 131)	*p*-Value
Time to first CRP level (median (IQR))	2.0 (8.3) ^a^	9.0 (16.5) ^a^	4.0 (15.0) ^b^	**0.013**
First CRP level (median (IQR))	1.0 (0.0)	10.5 (42.0) ^c^	1.0 (6.5) ^c^	**0.000**
Time to second CRP level (median (IQR))	32.0 (21.0) ^d^	32.0 (27.5) ^e^	32.0 (24.0) ^f^	0.922
Second CRP level (median (IQR))	2.0 (6.0) ^g^	24.0 (43.7) ^b^	5.0 (21.5) ^h^	**0.000**
Time to start of antibiotic therapy (median (IQR))	2.0 (4.0) ^i^	7.5 (19.3) ^j^	3.0 (12.0) ^b^	**0.010**

Bold *p*-Values are considered statistically significant. Abbreviations: CRP, C-reactive protein; IQR, interquartile range; mg/dL, milligrams per deciliter. ^a^ 2 missing, ^b^ 4 missing, ^c^ 1 missing, ^d^ 15 missing, ^e^ 5 missing, ^f^ 20 missing, ^g^ 14 missing, ^h^ 18 missing, ^i^ 3 missing, ^j^ 1 missing.

**Table 5 antibiotics-13-00388-t005:** Risks and recommendations according to the EOS calculator for neonates with a gestational age of ≥34 weeks in whom antibiotic therapy was discontinued or prolonged after 72 h.

	Discontinued AB (*n* = 55)	Prolonged AB (*n* = 36)	Total (*n* = 91)	*p*-Value
EOS risk at birth (mean (SD))	5.11 (11.3)	1.60 (2.7)	3.72 (9.1)	0.069
EOS risk after first clinical examination (mean (SD))	33.53 (87.5)	10.38 (20.8)	24.37 (69.9)	0.123
Recommended AB after first clinical examination (n (%))	39 (70.9)	17 (47.2)	56 (61.5)	**0.023**
Change of EOS risk during clinical course (n (%))	14 (25.5)	21 (58.3)	35 (38.5)	**0.002**
EOS risk during clinical course (mean (SD))	38.69 (89.5)	17.22 (21.9)	30.19 (71.5)	0.162
Change of EOS recommendation during clinical course (n (%))	13 (23.6)	17 (47.2)	30 (33.0)	**0.019**
Recommended AB after additional clinical examination (n (%))	52 (94.5)	34 (94.4)	86 (94.5)	0.984
EOS risk at start of AB (mean (SD))	36.85 (90.0)	13.31 (20.4)	27.54 (71.8)	0.127

Bold *p*-Values are considered statistically significant. Abbreviations: AB, antibiotics; EOS, early-onset sepsis; SD, standard deviation.

## Data Availability

Deidentified data will be shared upon request.

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
