# Peer review of "Factors Associated with Prolonged Antibiotic Therapy in Neonates with Suspected Early-Onset Sepsis"

_antibiotics, 2024, doi:10.3390/antibiotics13050388_

Round 1

Reviewer 1 Report

Comments and Suggestions for Authors

Dear Authors,

Interesting approach and findings.

I would like to make some comments to improve the presentation of your work.

You should add more bibliographic background to describe the problem that motivated the study.

In methodology I suggest you could indicate more clearly the inclusion and exclusion criteria used to develop the study.

It would be interesting if you could contrast your results with other similar studies (other than reference 13).

Kind regards

Reviewer 2 Report

Comments and Suggestions for Authors

Dear Authors,

the study is relevant and deserves attention. I did not find any major issues, apart from these mentioned below:

Introduction

In my opinion this section is too short and would benefit from expanding. I suggest to include some comments on the most common types of bacterial origin of EOS, most commonly administered antimicrobial agents in EOS cases and the most concerning outcomes of prolonged unnecessary antimicrobial therapy.

Conclusions section is vague to me. I suggest to rewrite this section to make it more clear to the reader what are the most obvious outcomes of the study and the suggestions that the Authors would have for future research directions.
